# Serum 25(OH)D Analysis in Captive Pachyderms (*Loxodonta africana*, *Elephas maximus*, *Diceros bicornis*, *Rhinoceros unicornis*, *Tapirus indicus*) in Europe

**DOI:** 10.3390/ani14192843

**Published:** 2024-10-02

**Authors:** Linda G. R. Bruins-van Sonsbeek, Ronald J. Corbee

**Affiliations:** 1Rotterdam Zoo, Blijdorplaan 8, 3041 JG Rotterdam, The Netherlands; 2Faculty of Veterinary Medicine, Department of Clinical Sciences, Utrecht University, Yalelaan 108, 3584 CM Utrecht, The Netherlands; r.j.corbee@uu.nl

**Keywords:** vitamin D, tapir, elephant, rhino, seasonal, latitude

## Abstract

**Simple Summary:**

This study aimed to determine the serum levels of vitamin D in captive pachyderms in Europe, and whether there were any seasonal differences. For some species, comparing the temperature and UVB irradiation of the current captive locations with the wild-ranging areas, the required conditions may be met in summer. In contrast, for other species, this might not be the case. Therefore, the animals may be unable to endogenously produce vitamin D, leading to calcium and phosphorus imbalances that were only reported in captive pachyderms. For some of the pachyderms, it is known that they can absorb cholecalciferol from the digestive tract, but this is not the case for all studied pachyderms. Tapirs had very low vitamin D levels, similar to horses. The higher vitamin D levels of elephants and rhinoceroses could indicate that they are capable of producing vitamin D or absorb more cholecalciferol from their diet. When comparing both rhinoceros and elephant species in this study, differences with regard to their capability in endogenous production of vitamin D were shown. Indian rhinoceroses seem capable of producing enough endogenous vitamin D year-round, while both elephant species and the black rhinoceros are not.

**Abstract:**

This study aimed to detect seasonal and species differences in serum 25-hydroxy vitamin D (25(OH)D) concentrations during summer and winter months in captive pachyderms in Europe. Both elephant species had low 25(OH)D while African elephants did not show a seasonal variation. Asian elephants had significantly higher 25(OH)D compared to their African counterparts but also did not show a seasonal difference. Both rhinoceros species investigated had higher 25(OH)D compared to both elephant species; the Indian rhinoceros had high circulating levels year-round, while the black rhinoceroses showed significantly lower 25(OH)D in winter. Malayan tapirs have very low 25(OH)D, comparable to horses. The higher 25(OH)D of elephants and rhinoceroses could indicate that elephants and rhinoceroses are capable of producing vitamin D. This might indicate that the Indian rhinoceroses are capable of producing enough endogenous vitamin D year-round at latitudes around 52° N, while both elephant species and the black rhinoceros are not. This study also showed that it is likely that both elephant species and rhinoceros species are capable of absorbing cholecalciferol from the digestive tract, according to the existing literature, while tapirs may not.

## 1. Introduction

Elephants, rhinoceroses, and tapirs, also commonly known as pachyderms, are often kept in zoos all over the world, predominantly for conservation purposes. Most of them are housed in institutions in Europe and North America at much higher latitudes than their natural range. Therefore, there are many differences regarding the natural habitat, the UVB radiation, and the environmental temperature of the captive versus wild pachyderms. This might contribute to hypovitaminosis D in captivity since both factors are crucial for endogenous vitamin D production, which can result in hypocalcemia and/or hypophosphatemia in humans [1] and might also lead to hypocalcemia and/or hypophosphatemia in captive pachyderms. In captivity, problems with hypocalcemia, especially in relation to dystocia and musculoskeletal problems, have been described for (captive) Asian elephants [2,3,4,5,6]. In captive Asian elephants, both in Europe and current range countries, hypocalcemia is reported [2,6,7], associated with unbalanced nutrition/too low dietary calcium (Ca) levels [6,8,9]. One study [7] demonstrated a significant increase in Ca plasma levels after oral Ca supplementation in summer, and oral cholecalciferol administration in summer and winter, in captive Asian elephants in Europe, but this did not occur in African elephants. To the author’s best knowledge, no reports are available on hypocalcemia in rhinoceroses or tapirs.

Based on the habitats and anatomy of the two elephant species, it is expected that serum 25(OH)D levels follow the same trend as the UV index, meaning it fluctuates with seasons, having a high summer 25(OH)D level and lower winter vitamin D level. However, they are often compared to horses in terms of their vitamin D metabolism [7,10,11]. If they are indeed comparable with horses, serum 25(OH)D levels would be low or undetectable [12]. It was also suggested that African elephants might have a higher threshold to endogenously produce cholecalciferol in the skin compared with Asian elephants, as seen in humans living around the equator versus people living in the Northern Hemisphere [7]. Both temperature and UV radiation in Europe are much lower compared to Africa (see Figure 1), so it might also be possible that the threshold to endogenously produce cholecalciferol is not met in African elephants and that African elephants will not show a seasonal trend and low overall 25(OH)D values in their serum. This condition is not only the case for the African species but for all pachyderms.

This study aimed to detect seasonal and species differences in the vitamin D levels during summer and winter months in Asian and African elephants, Malayan tapirs, and black and Indian rhinoceroses kept in captivity in (Western) Europe.

## 2. Materials and Methods

### 2.1. Samples

Serum collected for health monitoring of the captive pachyderms from the Rotterdam Zoo and stored at the biobank at the Rotterdam Zoo (−80 °C), and serum sent to the Rotterdam Zoo for progesterone analyses from pachyderms at other institutions from January 2020 to October 2023, were used for this study, with permission obtained for use of the samples. Details of individuals participating in the study can be found in Table 1.

For the African elephant, 97 unique samples of ten individual elephants housed at two different zoological institutions (Wuppertal Zoo, Hilvarenbeek Zoo) were available: age at the start of the monitoring period ranged from 1 to 35 years; eight females (age 4–35 years) and two males (age 1–27 years). For the Asian elephant, 94 unique samples of five individual elephants housed at one institution (Rotterdam Zoo; age 0–50 years old) were analyzed; four females (10–50 years) and 1 male (1 year). For the black rhinoceros, 90 unique samples of five individuals were available: age 0–19 years, of which three males (0–19 years) and two females (3–9 years) housed at one institution (Rotterdam Zoo), while for the Indian rhinoceros, four unique samples of one female rhinoceros housed at one institution (Rotterdam Zoo), five years of age at the start of the period (only samples from 2022 and 2023 were used) were used. For the Malayan tapir, 36 unique samples of two individual tapirs, aged 7–9 years; one male (seven years, only samples in 2023 were used) and one female (nine years) housed at one institution (Rotterdam Zoo) were analyzed.

All animals in all institutions had access to outside during summer for at least 8 h during the daytime. In winter this was often reduced to a maximum of 6 h during the daytime, depending on the temperature. No analyses were performed on the food items. Cholecalciferol intake was estimated based on the body weight of the animal, the amount of concentrates, and information on the amount of cholecalciferol on the package of the concentrates, which can be found in Table 1. Cholecalciferol supplementation was given at the same level during the whole year. Roughage was fed both inside and outside.

### 2.2. Vitamin D Analysis

Total serum 25(OH)D concentrations were determined by the VIDAS^®^ enzyme-linked fluorescent assay (ELFA) (Biomérieux, Marcy-L’Etoile, France). The VIDAS 25 OH vitamin D TOTAL assay combines an enzyme immunoassay competition method with a final fluorescent detection (ELFA) [19] with a detection limit from 20.3 to 315 nmol/L.

Based on the UV index [14], summer and winter can be divided into different months. A UV index of 1–2 means no UV radiation, a UV index of 3–4 represents very poor UV radiation, 5–6 a weak UV radiation, 7–8 is strong UV radiation, and 9–10 and higher represents a very strong UV radiation while >11 is extreme UV radiation. When looking at the months in the years 2020–2023, January, November, and December had a mean index of 0–1; May to July were graded 6–7. March and October always showed an index of two, while March, April, August, and September varied between three and five. Summer and winter were divided based on the UV index (<2 and >5, respectively) resulting in the winter being November, December, and January, and the summer being May, June, and July. On the 21st of March, spring officially starts (longer and more sunny days in Western Europe) and ends the 21st of June when summer begins. On the 21st of September, autumn officially starts (shorter and colder and cloudier days) and ends on the 21st of December, when winter starts. Here, we define summer as the period from April to September, and winter from October to March.

### 2.3. Statistical Methods

All tests were performed with R statistics (version 4.3.3). A Kolmogorov–Smirnov test was performed on all data of the African elephants, Asian elephants, and black rhinoceroses. None of the animals had normally distributed data; therefore, a Kruskal–Wallace rank sum test was performed to determine statistical differences between seasons in all species. As a post hoc test, a Dunn test was performed to determine the significant differences between the groups, with Holm–Bonferroni correction (ggstatsplot package in R).

To determine whether there was a significant difference between African and Asian elephants, a Wilcoxon rank sum test was performed. Alpha error in the analysis was set at *p* < 0.05. Some individuals accounted for more samples compared to others and, to correct for this, the mean was analyzed per animal per season, which was then counted for n = 1. The total mean was analyzed by calculating the mean per individual, which accounted for n = 1. The animals that contributed only with one or two samples or did not have samples in all seasons were only used to calculate the total mean and median (individual numbers 5, 6, 7, 10, 15, 20; Table 1) but were not used in the summer/winter comparison.

When serum levels were below the detection level of 20.3 nmol/L, we used 20.3 nmol/L in the statistics (see also sub-script of Table 2). While this might not be completely correct, the authors realize this might be more accurate than using 0.0 nmol/L. Since we have a limited sample size, we would not like to exclude those values.

No further testing was performed on the Indian rhinoceros, due to the low number of samples, and on the Malayan tapir who turned out to have numerous samples for which values were under the detection level.

## 3. Results

As mentioned earlier, the data for both elephants and the black rhinoceroses were normally distributed. Therefore, next to the total mean and standard deviation, the median and range are provided for all species in Table 2. The median and range of summer and winter based on the two different division techniques (UV index vs. calendar) are also shown in Table 2. Figure 1 shows the 25(OH)D analyses in all African and Asian elephants and black rhinoceroses over time. Figure 2 is a ggplot, showing the distribution of the samples during the different seasons of the African and Asian elephants and black rhinoceroses.

No significant differences could be detected between the two seasons for African elephants (*p* = 0.71) and Asian elephants (*p* = 0.09) by using the UV radiation index, nor when using the calendar. Comparing the mean of the Asian and African elephants, the latter showed a significantly lower 25(OH)D serum level (*p* < 0.001).

For black rhinoceroses, a significant difference in 25(OH)D in winter and summer was observed when the division was made based on both UV index and calendar data (index: *p* < 0.001; calendar: *p* < 0.001). The Indian rhinoceros data only contained four samples of one individual and only one of those samples originated in winter, which had a high concentration compared with the black rhinoceroses.

The Malayan tapir 25(OH)D levels were almost all under the detection level of 20.3 nmol/L.

For the Indian rhinoceros and Malayan tapir, no additional statistics were performed.

## 4. Discussion

The samples originate only from a small number of individuals housed in just 1–2 institutions. Due to multiple missing samples and to reduce the impact of individuals on the mean, a mean was calculated per season per animal or a mean per animal, reducing the sample size even further. The UV index in range countries is generally higher, except for the Asian elephant and the Indian rhinoceros, as shown in Figure 1. Therefore, it might not be a surprise that the African elephant serum 25(OH)D levels were significantly lower than those of their Asian counterparts, which supports the theory that the threshold for vitamin D production has not been met. However, another African megaherbivore, the black rhinoceros, did show significantly higher levels of serum 25(OH)D in summer despite the fact that it also lives in areas with a high UV radiation index and high ambient temperature, indicating that it might (at least partly) meet its threshold to endogenously produce vitamin D. The Indian rhinoceros showed higher mean vitamin D levels compared to black rhinoceroses, although no statistics could be performed due to the low sample size. Therefore, it is unclear whether the difference is statistically significant, which would require testing multiple Indian rhinoceroses. Furthermore, the Indian rhinoceroses showed less variation in summer and winter compared to the black rhinoceroses, perhaps due to adaptations to their natural environment, where they live in range countries where UV radiation is not always very strong or extreme (Figure 1). Although the Malayan tapir also lives in a natural environment with high temperatures and high UV radiation, they mainly live in dense forests and have a more nocturnal lifestyle compared to the other pachyderms [20,21]. This may be an explanation for their very low vitamin D levels.

Figure 3 shows that the variation in the UV index group is less compared to the season group. Comparisons between seasons should thus be based on the UV index rather than on calendar months.

The 25(OH)D levels in this study are considerably higher compared to the findings in other studies performed by the author ([7,22]; Table 3). Perhaps this might be due to a different test procedure since the VIDAS analyzes total 25(OH)D instead of only D_3_ metabolites. Here, a mean 25(OH)D of 34.5 ± 9.0 nmol/L in African elephants was found versus a mean (± SD) 25(OH)D_3_ in earlier studies of 11 ± 5 nmol/L [20] and 15.6 ± 7.7 nmol/L [7], respectively. Both institutions provided dietary cholecalciferol below the recommendation; however, the recommendations do not discriminate between D_2_ and D_3_ and, likely, these animals would also have received vitamin D_2_ from their roughage. When comparing the level of cholecalciferol in both institutions, institution B (3.6 IU/kg BW) gave about half the dosage of cholecalciferol compared to institution A (6.5 IU/kg BW), but no differences were found between the levels of 25(OH)D between both institutions (institution A median 34.4 (21.0–43.8 nmol/L), institution B median 30.8 (<20.3–71.5 nmol/L)).

A study conducted in Florida, North America in African elephants [11] found 25(OH)D levels of 39.4 ± 18.7 nmol/L, which are more comparable to the values in the present study, despite the difference in latitude between the Netherlands (52° N) and Florida (28° N). Vitamin D metabolism in Asian elephants has been more thoroughly investigated compared to their African counterparts. The mean 25(OH)D of Asian elephants reported here was 75.3 ± 21.1 nmol/L, versus 25(OH)D_3_ in earlier studies of 36 ± 11 nmol/L [22] and 35.6 ± 11.7 nmol/L [7]. This finding of much higher circulating levels of 25(OH)D compared to 25(OH)D_3_ in previous studies could indicate that both elephant species have high levels of circulating 25(OH)D_2_, which is supported by earlier studies [10,22,31]. Additionally, this can indicate that (Asian) elephants can absorb cholecalciferol added to their diet, which has been proven in Asian elephants in a recent study [31]. In earlier studies [10,31], captive Asian elephant sera samples were analyzed by discriminating 25(OH)D_2_ and D_3_ (43° N). They showed no detectable 25(OH)D_3_ levels in sera, and 25(OH)D_2_ levels were detected and were on average of 17.5 ± 2.2 nmol/L, which is much lower compared to the previous studies [7,22] and values reported here. A recent study analyzed the 25(OH)D levels in the serum of captive Asian elephants in India (Tamil Nadu; latitude 8–14° N). They found a mean of 27.7 ± 4.9 nmol/L (range: 21.2–35.4 nmol/L) [23], which is considerably lower compared to our findings and also compared to previous studies [7,22]. Except for the earlier studies [10,28,31], unfortunately, no details were provided on the diet or sunlight exposure [26] except that this was found to be sufficient. As mentioned above, this can be due to the use of total versus differentiated 25(OH)D tests, or differences in cholecalciferol supplementation/daily vitamin D intake. The current study was unable to demonstrate significant seasonal differences in circulating 25(OH)D levels in both elephant species, which is in comparison to previously mentioned studies [7,10,11,22,28]. However, in a previous study [7], a significant decrease in 1,25(OH)2D_3_ in African elephants was noticed during winter. This was negated when adding cholecalciferol to the diet, which indicates that African elephants might not be able to endogenously produce sufficient vitamin D in winter months in the Northern Hemisphere. In the same study in Asian elephants, a significant change in bone marker concentration, indicative of higher levels of bone resorption during the winter time, was found, which might also suggest that Asian elephants do not seem capable of producing sufficient amounts of cholecalciferol in winter months at the latitude of the study in the Northern Hemisphere and that Ca absorption might be (at least partly) dependent on vitamin D. However, it should be mentioned that it is still unknown whether both elephant species are capable of producing endogenous cholecalciferol at all.

In a previous study [23], a seasonal variation in 25(OH)D levels was detected in two captive Eastern black rhinoceros in America with the mean serum 25(OH)D level of 100.6 nmol/L. This is slightly lower compared to the mean of 126.2 ± 70.2 nmol/L reported here. However, our median is much lower, namely 86.3 nmol/L (range 72.9–237.0 nmol/L). A study in wild black rhinoceroses shows higher levels: a mean of 139.0 nmol/L [25] which is higher compared to our summer median value of 109.0 nmol/L (range 51.5–251.0 nmol/L) and winter median 58.3 nmol/L (range 23.5–226.0 nmol/L), which might indicate that some black rhinoceroses are not capable of producing enough endogenous vitamin D. However, these data should be interpreted with care, since the data include D_2_ and D_3_ and, like for the elephants, it is currently unknown whether they are capable of producing endogenous cholecalciferol. However, an earlier study [25] analyzed 25(OH)D_3_ in wild black rhinoceroses, in which it is highly unlikely they receive oral cholecalciferol. This makes it very plausible that black rhinoceroses are capable of endogenous cholecalciferol production. Horses, which are also a member of the Perissodactyla, like rhinoceroses and tapirs, depend on vitamin D_2_ in their diet and are not able to endogenously produce vitamin D_3_ [12,32,33]. However, they also show seasonal variation in 25(OH)D levels, which might be caused by the plants they consume, due to increasing levels of D_2_ due to the UVB radiation of the roughage [12]. Additionally, the black rhinoceroses received a different amount of cholecalciferol during the study, due to the change in type and amount of pellets they received. This could be the reason why the levels of 25(OH)D in these animals are higher in 2023 compared to the previous years (Figure 1). This finding makes it also very plausible that black rhinoceroses are capable of absorbing cholecalciferol in their digestive tract. This is supported by a previous study [23], which demonstrated a correlation between levels of cholecalciferol in the diet and plasma 25(OH)D levels in captive black rhinoceroses.

To the best of the author’s knowledge, no information is available on serum vitamin D levels in wild Indian rhinoceroses. There is one study from North America in captive Indian rhinoceroses which shows a significant difference in seasons regarding 25(OH)D levels [27]. Serum 25(OH)D in that study [27] ranged from 22.4 (± 2.93) in winter to 32.8 (± 7.44) nmol/L in summer, which is much lower compared to our findings of 106.3–132.8 nmol/L. An explanation for the high levels of 25(OH)D in the present study could be that the animals in that study [27] did not receive any cholecalciferol in their diet, which could indicate that Indian rhinoceroses are capable of absorbing cholecalciferol in their diet. The significantly higher levels of 25(OH)D in summer found [27] might indicate that Indian rhinoceroses are capable of endogenously producing vitamin D; however, it could also be caused by increasing levels of vitamin D_2_ in their diet as mentioned above.

If both Asian elephant and rhinoceros species show higher 25(OH)D levels compared to the African species, this could suggest that the Malayan tapirs should also have high values in summer, or overall levels of serum 25(OH)D as they share the same natural habitat. However, Malayan tapirs show very low vitamin D levels overall. Perhaps this is due to the fact that they are mainly nocturnal and are housed inside (out of fear of sunburn) or hide in the shade on very sunny days [34]. Some other nocturnal species like the Egyptian fruit bat also show undetectable levels of 25(OH)D in both captive and wild specimens [35]. However, when actively exposed to sunlight, they can increase their vitamin D levels, indicating that they can produce cholecalciferol in their skin [36]. Or perhaps their vitamin D physiology resembles that of horses, where the main vitamin D source is vitamin D_2_ and not cholecalciferol production in the skin [32]. This might also be the case for African elephants, which have very low circulating serum levels of 25(OH)D.

Comparing the very low serum 25(OH)D levels of the African elephants and Malayan tapirs with the intake of cholecalciferol (Table 1) might indicate that they either do not resorb cholecalciferol from the digestive tract or that cholecalciferol is not their main metabolite in vitamin D metabolism. This would support the hypothesis that they have a similar vitamin D metabolism to horses. Another option could be that the amount of cholecalciferol given was too low for these species, at least in all African elephants, and one of the Malayan tapirs, which were given lower amounts than current recommendations (Table 1). Also, Asian elephants in the study here received a much lower amount of cholecalciferol than recommended. However, it should be noted that this interpretation might not be correct (lower than the recommendations), since it is unknown what amount of ergosterol the animals received in their diet, and current recommendations are based on vitamin D and not cholecalciferol solely. As described above, the levels of 25(OH)D in this study were much higher than in previous studies and in the captive population in Asia.

Future research should be conducted to gain further insight into normal vitamin D metabolism; for instance, in vivo and in vitro testing of UVB radiation would be highly recommended by the author to investigate the capacity of producing endogenous vitamin D, especially since the UV index in range countries is much higher compared to places where most animals live in captivity (Europe and North America). However, it is known that (Malayan) tapirs are very sensible to sunburn, so in vivo testing for these animals needs to be conducted with great care. There are virtually no referential values of serum 25(OH)D, preferably with discrimination of 25(OH)D_2_ and 25(OH)D_3_ in these species in the wild, except for the black rhinoceroses. The presence of circulating 25(OH)D_3_ levels in wild animals might especially indicate whether these animals are capable of endogenous vitamin D production. Additionally, differentiation between pregnant animals, lactating animals, age and sex, time spent outside, and the amount of vitamin D (both cholecalciferol and ergosterol) in their diet to investigate the effect of those variables on serum 25(OH)D should be investigated.

## 5. Conclusions

African elephants have significantly lower serum 25(OH)D levels compared to their Asian counterparts when held in zoological institutions in the Northern Hemisphere. Circulating serum 25(OH)D levels in Asian elephants were higher compared to previous studies despite low levels of cholecalciferol in their diet. Both elephant species have low circulating serum 25(OH)D levels compared to the rhinoceroses in this study and do not show a seasonal variation. This might indicate that they are incapable of producing sufficient endogenous vitamin D year-round, or part of the year, or they receive insufficient vitamin D in their diet.

Both rhinoceros species showed much higher levels of 25(OH)D compared to all other pachyderms, which makes it more likely that they can endogenously produce cholecalciferol. The Indian rhinoceros had high circulating levels year-round, while the black rhinoceroses showed a significantly lower 25(OH)D level in the winter. This indicates that Indian rhinoceroses are capable of endogenously producing enough vitamin D year-round, while black rhinoceros are not.

Malayan tapirs have very low circulating levels of serum 25(OH)D, which might be due to their inability to endogenously produce cholecalciferol, an insufficient amount of vitamin D offered in their diet, or might be normal for this species.

## Figures and Tables

**Figure 1 animals-14-02843-f001:**
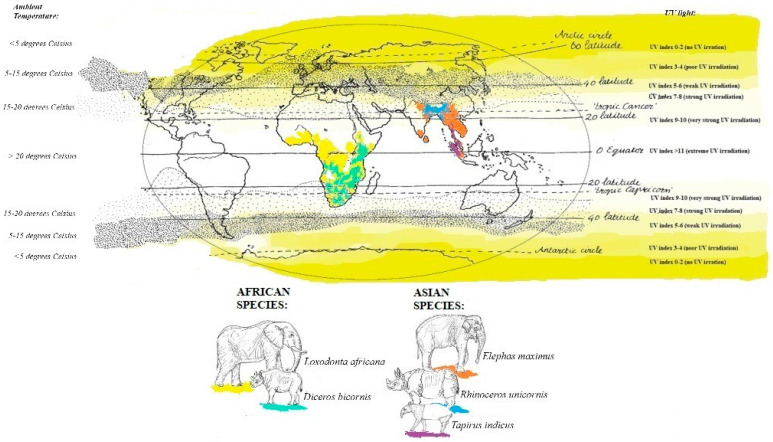
Distribution in the natural range on a map with UV index and temperature. The African elephant is in yellow, the black rhinoceros in light green, the Asian elephant in orange, the Indian rhinoceros in blue, and the Malayan tapir in purple [13,14,15].

**Figure 2 animals-14-02843-f002:**
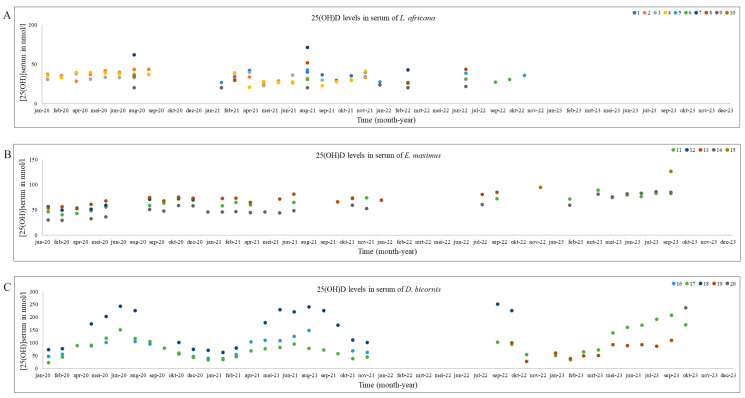
Serum 25(OH)D concentrations in nmol/L from individual African elephant (**A**), Asian elephant (**B**), and black rhinoceros (**C**) during 2020–2023. The numbers of the individuals in the legend correspond with the individuals mentioned in Table 1.

**Figure 3 animals-14-02843-f003:**
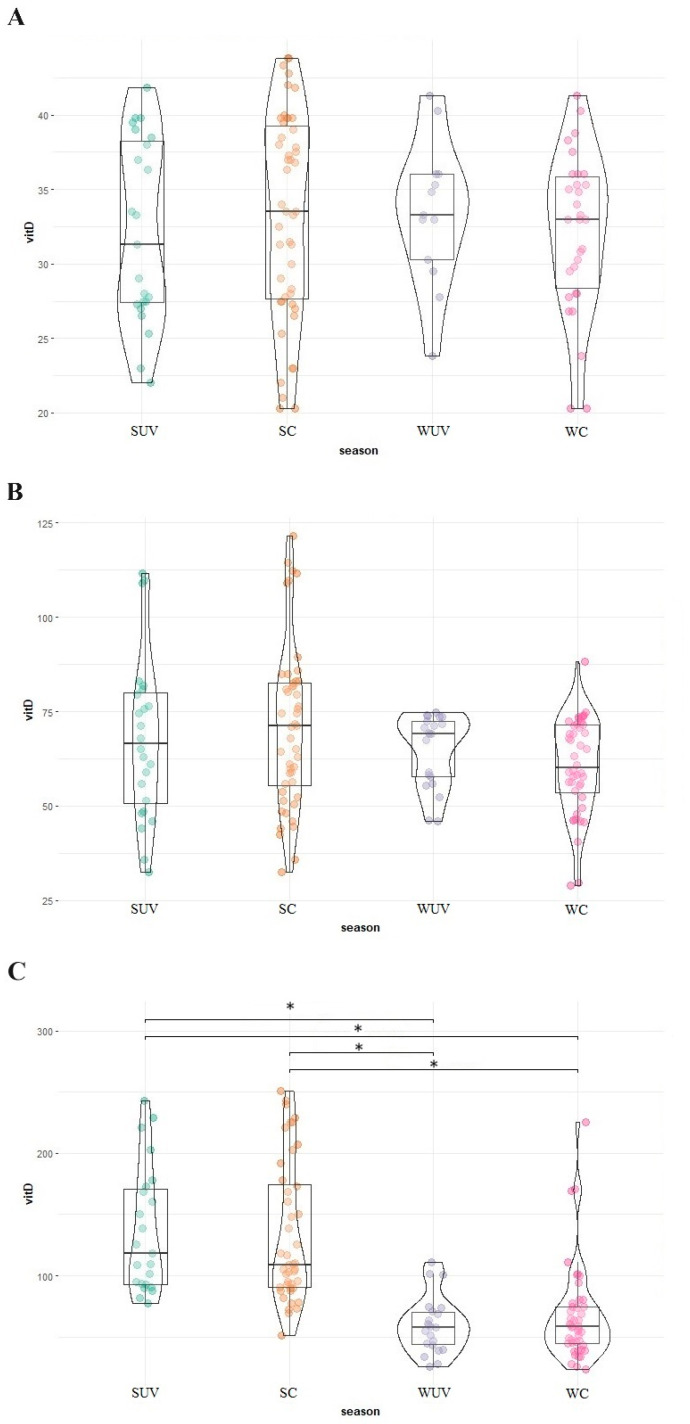
Distribution of serum 25(OH)D concentrations (nmol/L) in African elephants (**A**), Asian elephants (**B**), and black rhinoceros (**C**) per season. The bars with * indicate the significant difference between summer and winter values. Suv is summer according to UV index (green), sc is summer according to calendar months (orange), wuv is winter according to UV index (purple), and wc is winter according to calendar months (pink); vitD is 25(OH)D.

**Table 1 animals-14-02843-t001:** Participating institutions (Zoo), animal data, and calculated cholecalciferol intake of the animals through provided pellets.

Zoo *	Individual	Species	Sex	Year of Birth	Age at Sample Collection (Years)	n	Cholecalciferol Intake (IU/kg BW) **	Recommended Vitamin D Intake (IU/kg BW)
A	1	*L. africana*	f	2015	5–6	9	6.5	8–12^1^
	2	*L. africana*	f	1985	35–36	18	6.5	
	3	*L. africana*	f	1986	34–35	18	6.5	
	4	*L. africana*	f	1992	28–29	19	6.5	
B	5	*L. africana*	f	1993	27–29	6	3.6	
	6	*L. africana*	f	2007	13–15	3	3.6	
	7	*L. africana*	f	2016	4–6	5	3.6	
	8	*L. africana*	m	2019	2–3	6	3.6	
	9	*L. africana*	m	1993	34–36	5	3.6	
	10	*L. africana*	f	1992	29–31	1	3.6	
C	11	*E. maximus*	f	1970	50–53	25	5	12–15^1^
	12	*E. maximus*	f	2000	20–23	22	5	
	13	*E. maximus*	f	2003	17–19	20	5	
	14	*E. maximus*	f	2010	10–13	25	5	
	15	*E. maximus*	m	2021	1–2	2	5	
	16	*D. bicornis*	m	2001	19–20	19	2.5–6.6 ***	3–9^2^
	17	*D. bicornis*	f	2011	9–12	37	2.5–6.6 ***	
	18	*D. bicornis*	f	2017	3–5	22	2.5–6.6 ***	
	19	*D. bicornis*	m	2020	2–3	11	2.5–6.6 ***	
	20	*D. bicornis*	m	2019	4	1	2.5–6.6 ***	
	21	*R. unicornis*	f	2017	5–6	4	1.5	-
	22	*T. indicus*	f	2011	9–10, 12	32	10–30 ****	10.5^3^
	23	*T. indicus*	m	2016	7	8	5	

* Corresponding Zoos: (A) Hilvarenbeek Zoo; (B) Wuppertal Zoo; (C) Rotterdam Zoo. ** Cholecalciferol was present in concentrates (pellets or bricks), and intake was estimated based on the amount on the package (in international units (IUs)), the amount of pellets/bricks given, and the body weight (BW) of the animals. *** At the beginning of 2022, the type and amount of pellets had changed, with 6.6 IU/kg BW being the maximum they received when eating all pellets offered per day, which was not always the case. **** In June 2021, the diet was changed to reduce the BCS of the female, the concentrates changed from horse to rhinoceros pellets and the amount of pellets was reduced by 50%; therefore, she started with an amount of 30 IU/kg BW which evolved to 10 IU/kg BW in 2023. Recommendations: ^1^[16] ^2^[17] ^3^[18]; to the best of the author’s knowledge, there are no current recommendations for cholecalciferol in Indian rhinoceroses.

**Table 2 animals-14-02843-t002:** Serum 25(OH)D concentration (nmol/L) in captive pachyderms in Western Europe.

Species ***	Mean (±SD)	Median (Range)	Median Per Season (Range)				
			Summer			Winter	
			Calendar (Apr–Sept)	UV Index > 5 (May–July)		Calendar (Oct–Mar)	UV Index < 2 (Nov–Jan)
*L. Africana* * n = 90, 10 ^+^	34.5 (±9.0)	33.2 (21.2–58.8)	33.5 n = 47, 6 (20.3–43.8)	31.3 n = 23, 6 (22.0–41.8)		33.0 n = 30, 6 (20.3–41.3)	33.3 n = 13, 6 (23.8–41.3)
*E. maximus* n = 94, 5	75.3 (±21.1)	68.7 (55.5–110.6)	71.2 n = 48, 4 (32.5–122.0)	66.5 n = 24, 4 (32.5–112.0)		60.2 n = 44, 4 (29.0–88.3)	69.0 n = 21, 4 (46.0–74.8)
*D. bicornis* n = 90, 5	126.2 (±70.2)	86.3 (72.9–237.0)	109.0 † n = 44, 4 (51.5–251.0)	118.0 † n = 23, 4 (77.5–243.0)		58.3 n = 45, 4 (23.5–226.0)	57.9 n = 22, 4 (26.0–111.0)
*R. unicornis* n = 4, 1	-	107.9 (106.3–132.8)	109.5 (105.3–132.8)	121.2 (109.5–132.8)		106.3	106.3
*T. indicus* ** n = 26, 2	-	<20.3 (<20.3–33.5)	<20.3 (<20.3–< 20.3)	<20.3 (<20.3–< 20.3)		<20.3 (<20.3–33.5)	<20.3 (<20.3–33.5)

^+^ First value is the number of values, second values the number of individuals. * Four values were below 20.3 nmol/L. ** Many values were below 20.3 nmol/L, for the statistics 20.3 nmol/L was used as the actual value, but this is most likely inaccurate. *** Mean per individual was used to calculate the total mean and median; in Table 1, the number of samples provided per individual are reported. For the seasons, only the individuals with a complete dataset were used. † value is statistically different compared to the corresponding winter value.

**Table 3 animals-14-02843-t003:** Serum 25(OH)D, 25(OH)D_3_, and D_2_ (in nmol/L) analyses in pachyderms.

Species	Continent *	25(OH)D	25(OH)D_3_		25(OH)D_2_
Horse			0.1−6^9^ undetectable n = 21; 40° S^10^		<1.9−36.0^9^ 5−20 n = 21; 40° S^10^
		Captive	Captive	Wild	Captive
African elephant	Europe	34.5 ± 9.0 n = 90, 10 ^+^; 52° N^current study^	11± 5 n = 5; 52° N^12^ 15.6 ± 7.7 n = 6; 52° N^13^		

	North America	39.4 ± 18.7 n = 72, 14; 28° N^5^			
Asian elephant	Europe	75.3 ± 21.1 n = 90, 5; 52° N^current study^	36 ± 11 n = 8; 52° N^12^ 35.6 ± 11.7 n = 10; 52° N^13^		
	North America		undetectable n = 72, 6; 43° N^6,8^ undetectable n = 22; 28° N^8^ undetectable n = 5; 43° N^11^ 32.2 ± 8.7 ** n = 5; 43° N^11^		17.5 ± 2.2 n = 72, 6; 43° N^6,8^ 5.8 ± 1.5 n = 33; 28° N^8^ 12.3 ± 1.5 n = 5; 43° N^11^ 12.1 ± 2.2 ** n = 5; 43° N^11^
	Asia	27.7 ± 4.9 n = 10; 8−14° N^4^			
Black rhinoceros	Europe	† 126.2 ± 70.2 n = 90, 5; 52° N^current study^	0.24 ± 0 n = 2^2^	139.0 ± 85 n = 28; 20° S^3^	
	North America	† 100.6 n = 40, 2; 41° N^1^			

Indian rhinoceros	Europe	107.9 (106.3−132.8) n = 4, 1; 52° N^current study^			
	North America	† 27.6 (winter 22.4 (± 2.93) - summer 32.8 (± 7.44)) n = 10, 5; 40° N^7^			
Malayan tapir		< 20.3-33.5 n = 26, 2; 52° N^current study^			

^+^ First value is the number of values, second values the number of individuals. * Only for captive situation. ** After cholecalciferol supplementation. † Significant difference between summer and winter season. ^1–11^ Superscript numbers represent the following references: ^1^[23]; ^2^[24]; ^3^[25]; ^4^[26]; ^5^[11]; ^6^[10]; ^7^[27]; ^8^[28]; ^9^[29] (excluded the study on vitamin D toxicosis from this review); ^10^[30]; ^11^[31]; ^12^[22]; ^13^[7].

## Data Availability

The data presented in this study are available in the article.

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
