# Peer review of "Serum 25(OH)D Analysis in Captive Pachyderms (Loxodonta africana, Elephas maximus, Diceros bicornis, Rhinoceros unicornis, Tapirus indicus) in Europe"

_animals, 2024, doi:10.3390/ani14192843_

Round 1

Reviewer 1 Report

Comments and Suggestions for Authors

Overview of project

The authors took samples from the pachyderms at a couple of zoos, compared them to each other and over two seasons (determined by two methods).   Comparative data for range country values is available only for elephants, it is spotty for rhinos and unavailable for tapirs.

The simple summary seems overly simple as the goal is not detailed and it appears that the authors compared species to each other and then made conjectures about the physiology before comparing to normal vitamin D levels in their natural range- this does not seem like a comparison that would yield useful information.  In the scientific abstract there is a better explanation of the goals (detect seasonal and species differences), and it is explained very well in the introduction and discussion.  The simple summary can be improved to better reflect the project and thought process that is better explained in the rest of the manuscript

There is not enough information in the paper (or it is not well explained) why the authors are focusing on endogenous production rather than variations in dietary intake, or species differences in absorption, or species differences in the requirements.

The analysis is well explained and I was glad to see that the authors did account in the data analysis for the fact that some animals had more samples than others.

Comments about the UV index in range countries is correct except maybe for tapirs, while the UV index is higher in those countries, all tapir species are forest (thick forests often) dwelling animals that spend a lot of time in the water, thereby drastically reducing their sunlight exposure compared to the UV technically available in the area. It would improve the discussion to consider this.

The authors did a nice job compiling the available data in the literature, but since there are so many differences in what was measured, a table would really help the reader put the obtained data in the context of existing data (literature)

Line 123- the package of food? Supplements?

Table 2- it would be more useful to have the age range for the samples for each individual than the birth year

Figure 2 and 3 the font in the X-axis is too small to read

 My main comment is that the authors seem to focus on the ability of the animals to produce Vitamin D endogenously, but data on diet analysis to know what intake is, normal ranges in free-ranging animals, parathyroid hormone levels, data on how much is absorbed from the diet, is not included (and much of it is in fact non-existent in the literature) to provide context for this focus as opposed to other alternatives for the species differences.

Comments on the Quality of English Language

While the language is generally very good, there are also a number of small language errors, so it would be recommended to have a native speaker proofread the text. A couple of examples:

Line 33- the expression is “year round” (not around)

Line 34 SignificantLY lower levels, rather than significant lower levels

Line 346 Incapable, not uncapable

Author Response

Dear Reviewers,

Thank you so much for your time, comments and suggestions which improved the quality of the manuscript. See our responses in the text below.

Sincerely yours,

Linda Bruins-van Sonsbeek

Reviewer 1: Overview of project

The authors took samples from the pachyderms at a couple of zoos, compared them to each other and over two seasons (determined by two methods). Comparative data for range country values is available only for elephants, it is spotty for rhinos and unavailable for tapirs.

Comment 1: The simple summary seems overly simple as the goal is not detailed and it appears that the authors compared species to each other and then made conjectures about the physiology before comparing to normal vitamin D levels in their natural range- this does not seem like a comparison that would yield useful information.  In the scientific abstract there is a better explanation of the goals (detect seasonal and species differences), and it is explained very well in the introduction and discussion.  The simple summary can be improved to better reflect the project and thought process that is better explained in the rest of the manuscript

Response 1: thank you for your comment. The problem is that there are no good vitamin D reference ranges for these species in the wild. There has only been one paper in black rhino from 2002 and recently there has been released a new paper on captive Asian elephants in their natural range country, however this article only states that access to sunlight and diet were sufficient but did not go into detail. The summary has been adjusted: see in line 10-26, line 30-31 and line 34-38, extra information and differences in these sentences are written in red.

Comment 2: There is not enough information in the paper (or it is not well explained) why the authors are focusing on endogenous production rather than variations in dietary intake, or species differences in absorption, or species differences in the requirements.

Response 2: thank you for your comment. We have added in the abstract a sentence on the absorption of cholecalciferol absorption (57-59 in red). In the discussion and conclusion this was already mentioned for all species as far as we are aware.

The reason for the focus on the possibility to produce cholecalciferol is because the very clear differences in the natural environment. I do agree that the tapirs are forest dwellers and might not get so much UV light, in fact they also have a much more nocturnal lifestyle compared to the other pachyderms in this study. In the Malayan tapir there is not so much problems with mineral imbalances compared to the other pachyderms, however there are many, mainly infectious diseases described in this species compared to the wild, which might be due to housing or impaired immune response. Since it is known that vitamin D can also have an effect on the immune system, it was interesting to be able to study this in this species as well, furthermore they are more closely related to rhinoceroses than horses, so comparing them might perhaps give a better insight for the aetiology in for instance hypophosphatemia in the black rhino. (but of course much more study is needed and this cannot be concluded from this study, but we did find the results from this study worth sharing since it does share some new information especially with regards to the tapirs). There is not so much known on the requirements for this species and not so much information from the wild. In Asian elephants and Indian and black rhinoceroses it is known they can absorb cholecalciferol from there digestive tract. However, there are also some reports that for instance Asian elephants are more dependent on ergosterol rather than cholecalciferol. In black rhino there have also been some reports on hypervitaminoses D due to production failure in the pellets. Therefore, would focus more on the ability to endogenously produce vitamin D, since this is much better regulated I the body and less risk to develop a hypervitaminoses D. And in the future perhaps see if there are recommendations needed to hang UV lights instead of increasing amounts of vitamin D in the diet.   

Comment 3: The analysis is well explained and I was glad to see that the authors did account in the data analysis for the fact that some animals had more samples than others.

Response 3: Thank you ?

Comment 4: Comments about the UV index in range countries is correct except maybe for tapirs, while the UV index is higher in those countries, all tapir species are forest (thick forests often) dwelling animals that spend a lot of time in the water, thereby drastically reducing their sunlight exposure compared to the UV technically available in the area. It would improve the discussion to consider this.

Response 4: Thank you, you are correct, we have clarified this further in the discussion, line 239-243  in red

 Comment 5: The authors did a nice job compiling the available data in the literature, but since there are so many differences in what was measured, a table would really help the reader put the obtained data in the context of existing data (literature)

Response 5: thank you, a table has been added (table 3)

Comment 6: Line 123- the package of food? Supplements?

Response 6: adjusted it (now line 138) in red to package of the concentrates

Comment 7: Table 2- it would be more useful to have the age range for the samples for each individual than the birth year

Response 7: thank you for your suggestion, this is added to table 1

Comment 8: Figure 2 and 3 the font in the X-axis is too small to read

Response 8: Adjusted the size of the text in figure 3; is it possible to turn figure 2 90degrees? Is this than still considered to small? Thank you.

Comment 9: My main comment is that the authors seem to focus on the ability of the animals to produce Vitamin D endogenously, but data on diet analysis to know what intake is, normal ranges in free-ranging animals, parathyroid hormone levels, data on how much is absorbed from the diet, is not included (and much of it is in fact non-existent in the literature) to provide context for this focus as opposed to other alternatives for the species differences.

Response 9: thank you for your point, see also the response 2 for a further explanation. I totally agree that PTH would have been a very good hormone to also take into consideration, however, we have only analysed the vitamin D levels. Adding calcium (and also make a discrimination between total and ionized calcium) magnesium and phosphorus would also been ideal, and also other hormones like the addition of for instance FGF23. Hopefully in the future we will be able to analyse all these different hormones and minerals. While we did calculate how much cholecalciferol was in the diet, we did not analyse this, nor did we analyse the amount of ergosterol in the diet, which would be a very valuable addition. Especially in the case of seasonal differences. A discrimination between 25(OH)D2 and D3 in the serum as well.

Comment 10: Comments on the Quality of English Language. While the language is generally very good, there are also a number of small language errors, so it would be recommended to have a native speaker proofread the text. A couple of examples:

  • Line 33- the expression is “year round” (not around)
  • Line 34 SignificantLY lower levels, rather than significant lower levels
  • Line 346 Incapable, not uncapable

Response 10: thank you for your comments and suggestions.

Reviewer 2 Report

Comments and Suggestions for Authors

 The authors state that all the animals had access to going outside for 5 or 8 hours per day, but not whether the tested animals actually did go outside. This may make a difference to the amount of sunlight and Vit D available to them?

We have found with horses over many years, that if they are offered a mixture of minerals, including a great deal of calcium ( 70%) in  feeding lime, and when sunlight is low  with cod liver oil additive, we can solve foot problems which we previously had. These animals are given ad libitum access to this mix and use their "nutritional wisdom" to eat it when they want at the level they want since it does no harm if they eat too much. I wonder why this type of approach is not used in zoos?? The animals do have to learn to take it, but they do if it is left always in their enclosure. This I began about 50 years ago because of reading an article which had calculated the amount of Ca required for equines, and how this was not usually met, and it has worked so far. Just a thought. 

Comments on the Quality of English Language

English fine, some type probably usually are. 

Author Response

Dear Reviewers,

Thank you so much for your time, comments and suggestions which improved the quality of the manuscript. See our responses in the text below.

Sincerely yours,

Linda Bruins-van Sonsbeek

Reviewer 2:

Comment 1: The authors state that all the animals had access to going outside for 5 or 8 hours per day, but not whether the tested animals actually did go outside. This may make a difference to the amount of sunlight and Vit D available to them?

Response 1: Thank you for your question, we totally agree that this would make an enormous difference. In this case the animals were outside during that period, with the exception when there was very bad weather conditions (heavy storms, freezing outside etc, however, these conditions are not that often so bad the animals will be kept inside but they will have been outside for most of this period)

Comment 2: We have found with horses over many years, that if they are offered a mixture of minerals, including a great deal of calcium ( 70%) in  feeding lime, and when sunlight is low  with cod liver oil additive, we can solve foot problems which we previously had. These animals are given ad libitum access to this mix and use their "nutritional wisdom" to eat it when they want at the level they want since it does no harm if they eat too much. I wonder why this type of approach is not used in zoos?? The animals do have to learn to take it, but they do if it is left always in their enclosure. This I began about 50 years ago because of reading an article which had calculated the amount of Ca required for equines, and how this was not usually met, and it has worked so far. Just a thought. 

Response 2: thank you very much for sharing your experience.

Reviewer 3 Report

Comments and Suggestions for Authors

Overall, it is well written manuscript discussing the changes of serum vitamin D status in pachyderms. Although the sample size is limited, some of these numbers can still serves as a reference for future studies. However, here are a few suggestions:

Line 66-67: I question this conclusion that horses have low or undetectable levels of serum VD. The paper cited showed differences between 25OHD2 and 25OHD3. And others have shown D3 levels being pretty significant in horses. https://doi.org/10.1016/j.tvjl.2014.01.002

Line 87-162: Much more information should be provided in Materials and Methods, despite the fact that some of them were already in table and figure legends. For example, in Table 2, the authors explained how values<20.3 were adjusted. This should be elaborated in section 2.2 or 2.3 as well. The Materials and Methods portion should be detailed enough to cover the reasons for many of your decisions.

Line 108: Table 1, I suggested adding ages of animals in 2023 or Age when samples were taken as a range. It would be more quantitative then year of birth.

Line 121-122: To my knowledge, vitamin D status can be largely influenced by VD intake. In Table 1, animals in Zoo A and B had significantly different vitamin D intake. Any adjustments made when pooling these numbers? Any other information on their diets in general?

Line 129 and 403: I don’t think citation 13, the assay manual, is needed.

Line 137: Here summer and winter were divided by UV index <2 and >5, and in Table 2, they were >6 and <2, was this a typo or different numbers used?

Line 216-218: Maybe I missed this, but I think it is worth noting in Figure 3 B and C, the wuv group had less variation vs. wc group, which suggested your way of dividing seasons by UV index might be a better way to look at seasonal changes than calendar month.

Author Response

Dear Reviewers,

Thank you so much for your time, comments and suggestions which improved the quality of the manuscript. See our responses in the text below.

Sincerely yours,

Linda Bruins-van Sonsbeek

Reviewer 3:

Overall, it is well written manuscript discussing the changes of serum vitamin D status in pachyderms. Although the sample size is limited, some of these numbers can still serves as a reference for future studies. However, here are a few suggestions:

Comment 1: Line 66-67: I question this conclusion that horses have low or undetectable levels of serum VD. The paper cited showed differences between 25OHD2 and 25OHD3. And others have shown D3 levels being pretty significant in horses. https://doi.org/10.1016/j.tvjl.2014.01.002

Response 1: thank you for sharing this very valuable article. We were aware sometimes 25(OH)D3 levels too low to detect or low, but most often these studies had supplemented horses with cholecalciferol. I do see this article they were not, however they still conclude compared to all other species known they have very low levels of 25(OH)D3, so this would most likely not alter our conclusion.

Comment 2: Line 87-162: Much more information should be provided in Materials and Methods, despite the fact that some of them were already in table and figure legends. For example, in Table 2, the authors explained how values<20.3 were adjusted. This should be elaborated in section 2.2 or 2.3 as well. The Materials and Methods portion should be detailed enough to cover the reasons for many of your decisions.

Response 2: thank you for your comment, we have added this in the m and m as line 176-179 in red.

Comment 3: Line 108: Table 1, I suggested adding ages of animals in 2023 or Age when samples were taken as a range. It would be more quantitative then year of birth.

Response 3: thank you, we have adjusted this in the table, this was also suggested by reviewer one.

Comment 4: Line 121-122: To my knowledge, vitamin D status can be largely influenced by VD intake.

Response 4: we totally agree ?. We have added some clarification with regards to this in the m and m line 138-140 in red. This was also already part of the discussion.

Comment 5: In Table 1, animals in Zoo A and B had significantly different vitamin D intake. Any adjustments made when pooling these numbers? Any other information on their diets in general?

Response 5: we have added this point in the discussion, there we no differences between the institutions, line 253-259 in red. We have more details with regards to the diet, but this is all based on calculations from the package, but for instance calcium and phosphorus levels are known. Unfortunately no analyses were performed and was not possible anymore since we have analysed this retroactively.

Comment 6: Line 129 and 403: I don’t think citation 13, the assay manual, is needed.

Response 6: thank you for your remark.

Comment 7: Line 137: Here summer and winter were divided by UV index <2 and >5, and in Table 2, they were >6 and <2, was this a typo or different numbers used?

Response 7: thank you very much for this remark. It has to be > 5 (we only included 6 and higher), it is adjusted in the table in red.

Comment 8: Line 216-218: Maybe I missed this, but I think it is worth noting in Figure 3 B and C, the wuv group had less variation vs. wc group, which suggested your way of dividing seasons by UV index might be a better way to look at seasonal changes than calendar month.

Response 8: thank you for your comment, we have added this to the discussion

(line 244-247) in red.

Round 2

Reviewer 3 Report

Comments and Suggestions for Authors

All my comments were addressed, no further comments.